# Oxidative Stress Evaluation in Ischemia Reperfusion Models: Characteristics, Limits and Perspectives

**DOI:** 10.3390/ijms22052366

**Published:** 2021-02-27

**Authors:** Pauline Chazelas, Clara Steichen, Frédéric Favreau, Patrick Trouillas, Patrick Hannaert, Raphaël Thuillier, Sébastien Giraud, Thierry Hauet, Jérôme Guillard

**Affiliations:** 1Maintenance Myélinique et Neuropathies Périphériques, Université de Limoges, EA 6309, 87032 Limoges, France; pauline.chazelas@gmail.com (P.C.); frederic.favreau@unilim.fr (F.F.); 2Laboratoire de Biochimie et Génétique Moléculaire, CHU de Limoges, 87042 Limoges, France; 3INSERM U1082, IRTOMIT, 86021 Poitiers, France; clara.steichen@gmail.com (C.S.); patrick.hannaert@univ-poitiers.fr (P.H.); rathuillier@gmail.com (R.T.); giraudseb@yahoo.fr (S.G.); thierry.hauet@gmail.com (T.H.); 4Faculté de Médecine et de Pharmacie, Université de Poitiers, 86074 Poitiers, France; 5INSERM U1248, IPPRITT, Université de Limoges, 87032 Limoges, France; patrick.trouillas@unilim.fr; 6RCPTM, University Palacký of Olomouc, 771 47 Olomouc, Czech Republic; 7Service de Biochimie, CHU de Poitiers, 86021 Poitiers, France; 8FHU SUPORT Survival Optimization in Organ Transplantation, 86021 Poitiers, France; 9IBiSA Plateforme Modélisation Préclinique-Innovations Chirurgicale et Technologique (MOPICT), Do-maine Expérimental du Magneraud, 17700 Surgères, France; 10UMR CNRS 7285 IC2MP, Team 5 Chemistry, Université de Poitiers, 86073 Poitiers, France

**Keywords:** oxidative stress, Reactive Oxygen Species (ROS), antioxidant factors, ischemia-reperfusion injury, animal models, organoids, molecular modeling models

## Abstract

Ischemia reperfusion injury is a complex process consisting of a seemingly chaotic but actually organized and compartmentalized shutdown of cell function, of which oxidative stress is a key component. Studying oxidative stress, which results in an imbalance between reactive oxygen species (ROS) production and antioxidant defense activity, is a multi-faceted issue, particularly considering the double function of ROS, assuming roles as physiological intracellular signals and as mediators of cellular component damage. Herein, we propose a comprehensive overview of the tools available to explore oxidative stress, particularly in the study of ischemia reperfusion. Applying chemistry as well as biology, we present the different models currently developed to study oxidative stress, spanning the vitro and the silico, discussing the advantages and the drawbacks of each set-up, including the issues relating to the use of in vitro hypoxia as a surrogate for ischemia. Having identified the limitations of historical models, we shall study new paradigms, including the use of stem cell-derived organoids, as a bridge between the in vitro and the *in vivo* comprising 3D intercellular interactions *in vivo* and versatile pathway investigations in vitro. We shall conclude this review by distancing ourselves from “wet” biology and reviewing the in silico, computer-based, mathematical modeling, and numerical simulation options: (a) molecular modeling with quantum chemistry and molecular dynamic algorithms, which facilitates the study of molecule-to-molecule interactions, and the integration of a compound in a dynamic environment (the plasma membrane...); (b) integrative systemic models, which can include many facets of complex mechanisms such as oxidative stress or ischemia reperfusion and help to formulate integrated predictions and to enhance understanding of dynamic interaction between pathways.

## 1. Introduction

Oxidative stress (OS) is the consequence of an imbalance between the production of reactive oxygen/nitrogen species (ROS/RNS) and the organism’s capacity to counteract their action by the antioxidative systems. OS initiates structural and functional alterations of key biomolecules such as nucleic acids, lipids, and proteins, produces antioxidant defense system imbalance, and is involved in several pathophysiological mechanisms. OS is essential to the maintenance of biological homeostasis, including protection against pathogens and intracellular signaling across all organ systems, tissue, and subcellular structures in a highly compartmentalized but interconnected manner [1]. OS should consequently be considered as a two-sided entity, encompassing, on the one hand, the maintenance of low physiological levels of ROS, essential to the governing life processes through redox signaling (oxidative eustress), and, on the other hand, excessive and pathological levels of ROS, which cause OS-related damage [2]. Recently, in order to better characterize the complex interactions of ROS, the concept of reactive species interactome has been introduced [3]. In this review, we have focused on ischemia reperfusion injury (IRI) which is defined as the paradoxical exacerbation of cellular dysfunction and death, following restoration of blood flow to previously ischemic tissues. IRI occurs in a wide range of organs including the heart, lung, kidney, gut, skeletal muscle, and brain, and it may not only involve the ischemic organ itself, but also cause systemic damage to distant organs, potentially leading to multi-system organ failure [4,5]. Recent works have suggested that IRI occurs as a result of specific processes and is not restricted to a sudden and catastrophic breakdown of cell function [6,7]. IRI involves interlinked signaling pathways such as the NADPH oxidase system, the nitric oxide synthase system, and the xanthine oxidase system. In these processes and particularly during reperfusion, OS is associated with a network including inflammation and reticulum endoplasmic stress, among others. Based on recent findings on renal vasculature and cellular stress responses (primarily at the intersection of unfolded protein response), mitochondrial dysfunction, autophagy, and innate immune response have been reported [8]. It is well-established that IRI is one of the most common causes of acute kidney injury (AKI), and there is increasing recognition that acute kidney injury (AKI) and chronic kidney disease (CKD) are closely linked and likely promote one another [9]. Notwithstanding many studies, the mechanisms underlying IRI remain highly unclear and in vitro and *in vivo* models epitomizing the fundamental processes are of paramount interest in research on the pathogenesis of IRI and as means of highlighting consistent targets and plausible therapeutics.

In the first part of this review, we present the main biochemical aspects of ROS and OS. In the second part, we propose an overview of the various experimental models of OS, first in animals and in relation to IRI, and second in cell models and with regard to hypoxia reoxygenation (HR). In the third section, we address emerging concepts in the modelling of OS, starting with the promising generation of organoids, as a high-potential approach helping to unravel mechanisms, to identify specific targets and to validate specific, measurable relevant biomarkers, after which a brief survey of *in silico* approaches is given.

## 2. Biochemical Aspects of Oxidative Stress

### 2.1. Definition of Oxidative Stress

OS is a phenomenon originating from the uncontrolled production of reactive species, mainly oxygen or nitrogen. The key characteristic of these molecules is their high reactivity toward biological constituents, lipids, proteins, nucleic acids, or carbohydrates. Scheme 1 underlines the detrimental role of uncontrolled ROS production in deoxyribo nucleic acid (DNA) injury.

Here, we have focused on ROS as derivatives induced by partial reduction of dioxygen in cells or in the body. The formation of these compounds, particularly superoxide anion (O_2_°^−^), is linked to the transfer to dioxygen of electrons issued from different processes such as the dysfunction of oxidative phosphorylation, the recruitment of inflammatory cells, or the massive ROS-producing enzyme activation occurring in ischemia reperfusion injury [10,11].

### 2.2. Reactive Oxygen Species

In physiological conditions, ROS are produced in limited quantities by the normal cell oxidative metabolism of the cell, with about 2% of the O_2_ consumed by the mitochondrial respiratory chain transformed into ROS [12]. However, during OS, ROS are generated at much higher levels. Among ROS, the superoxide anion (O_2_°^−^) is one of the first produced. In turn, through chain reactions, O_2_°^−^ will lead to the formation of the hydroxyl radical (OH°), the perhydroxyl radical (HO_2_), the peroxyl radical (RO_2_°) and the alkoxyl radical (RO°). A free radical is a chemical species featuring a single electron in its valence orbital, which endows it with very strong reactivity towards surrounding molecules. The dismutation of O_2_°^−^ by superoxide dismutase (SOD) can also lead to the formation of reactive hydrogen peroxide (H_2_O_2_). While H_2_O_2_ is not a radical species per se, it exhibits high toxicity through reactions with ionic species, especially transition metals (Scheme 2). The toxicity of O_2_°^−^ results in: (1) its direct interactions with the various cellular constituents, (2) its chemical reactions at the origin of the ROS mentioned above, and finally (3) its reaction with NO to form reactive oxygenated nitrogenous species including peroxynitrite (ONO_2_^−^), which causes nitro-oxidant stress and vasomotor imbalance *in vivo*. With OH°, ONO_2_^−^ is the most reactive radical and induces significant cytotoxicity due to its numerous targets and cellular actions [13].

### 2.3. Sources of Reactive Oxygen Species

Among the different sources of ROS, the most widely studied are the mitochondrial respiratory chain, the NADPH oxidase, the xanthine oxidase, and the nitric oxide synthetases.

#### 2.3.1. The Mitochondrial Respiratory Chain

The mitochondrion is the primary source of O_2_°^−^ during IRI. During ischemia, when aerobic metabolism is slowed or stopped, reduced coenzymes (NADH,H^+^ and FADH_2_) accumulate since they cannot be recycled by mitochondrial metabolic activity. At the time of reperfusion, their sudden emergence generates a major release of electrons, usually exceeding the transfer capacity of the respiratory chain; some electrons are thereby transferred to O_2_ producing O_2_°^−^. This abnormal transfer of electrons takes place mainly at the levels of complexes I and III [14,15,16]. In addition, the alteration of respiratory chain compounds by ischemic acidosis and by ROS during reperfusion reinforces mitochondrial dysfunction, exacerbating the production of ROS by mitochondria.

#### 2.3.2. NADPH Oxidase

Nicotinamide adenine dinucleotide phosphate (NADPH) oxidase is a membrane enzyme that catalyzes the reduction of O_2_ to O_2_°^−^ using NADPH,H^+^ as a substrate. Activated during the inflammatory process and/or OS, phagocytic NADPH oxidase plays a major role in the production of ROS, which has been widely studied in cardiovascular diseases [17,18]. In cardiac IRI conditions, NADPH oxidase from endothelial cells and from cardiomyocytes has been repeatedly reported as a major source of ROS, thereby contributing to the severity of reperfusion injuries [19].

#### 2.3.3. The Xanthine Oxidase (XO) Pathway

After activation, XO is an enzyme that catalyzes the oxidation of hypoxanthine and xanthine in purine metabolism using O_2_. During ischemia reperfusion (IR), calcium overload and reduced ATP synthesis cause XO activation from xanthine dehydrogenase and ROS production. In physiological conditions, xanthine dehydrogenase is known to induce hypoxanthine and xanthine oxidation through NAD^+^ reduction [20,21].

#### 2.3.4. Nitric Oxide Synthetases (NOS)

NOS are metallo-enzymes which, in their homodimeric form, catalyze the synthesis of NO°. In pathological conditions, they may be present as monomers involved in higher O_2_^°−^ production, minimizing NO° synthesis [22].

### 2.4. Antioxidant Factors

Intra and extracellular production of reactive species is constantly controlled by a dense “antioxidant/redox defense network” involving different types of antioxidant factors (AO). Mechanistically, most of them act via one of the following antioxidant mechanisms (i) sequestration of transition metal ions, (ii) scavenging and quenching of reactive species, (iii) ending of chain reactions sustained by free radicals, and (iv) molecular repair of radical damage. All in all, several hundreds of simple and complex compounds are involved in antioxidant defense. It is beyond the scope of this review to make an exhaustive or detailed presentation of these actors. Here, we briefly screen and summarize them based on the “chemical” distinction that is usually made between the non-enzymatic, or “chemical” antioxidant (ChAO, Table 1) and the enzymatic antioxidant (EnAO, Table 2) systems.

#### 2.4.1. The Non-Enzymatic Antioxidant System

The ChAO factors can be further divided into hydrophilic antioxidant factors, operating in intra- and extra-cellular compartments, and lipophilic, “membrane-targeted” antioxidant factors. In addition, some factors are amphiphilic and can act in either polar or non-polar environments. Table 1 lists and comments on the main chemical antioxidant systems [23,24,25,26,27,28,29,30,31].

Numerous ChAO factors contribute to antioxidant defenses directly, i.e., by scavenging (one or several) reactive species, and also indirectly by: (i) maintaining or stabilizing complementary AO factors, (ii) repairing damage caused by reactive species to cell components (lipids and membranes, proteins, DNA), and/or (iii) modulating the expression of AO enzymes (see Table 2 below); e.g., via the Nrf-2 “redox master” transcription factor [25,32].

#### 2.4.2. The Enzymatic-Protein Antioxidant System

Even though the efficacy and therapeutic benefits of the non-enzymatic AO factors presented above are well-established [24,33], they usually remain less effective than the endogenous enzymatic AO factors [33]. Conversely, as primarily enzymatic compounds, they usually present high specificity toward a given reactive species (e.g., SOD vs. O_2_°^−^, or catalase vs. H_2_O_2_, see Scheme 1, below). The main components of the EnAO system are listed in Table 2.

Among the above, the most prominent, most widely studied and most frequently cited EnAO factors, are the following three enzymes: superoxide dismutase (SOD), glutathione peroxidase (GPx), and catalase (CAT) (Scheme 2).

Superoxide Dismutase (SOD)

It is a metallo-enzyme that catalyzes the dismutation of O_2_°^−^ into H_2_O_2_ and O_2_. Based on its front-line intervention in the elimination of O_2_°^−^, its role in the control of cellular redox status and survival in an aerobic environment is decisive. There are three different isoforms of SOD, which differ according to the metal involved in the holoenzymes and in their cellular location: copper-zinc SOD (Cu-Zn-SOD or SOD1) present in the cytoplasm and the mitochondrial inter-membrane space; manganese SOD (Mn-SOD or SOD2) localized in the mitochondrial matrix and inner membrane; and extracytosolic Cu-Zn-SOD (or SOD3), which is found mainly in the extracellular matrix. Since the activity of SOD produces H_2_O_2_, its antioxidant action has got to be backed up by H_2_O_2_ removing enzymes, GPx, and CAT (see below).

b.Glutathione peroxidase (GPx)

GPx eliminates H_2_O_2_ by coupling its reduction with the oxidation of a reducing substrate, glutathione: H_2_O_2_ is converted into H_2_O while reduced glutathione (GSH) is transformed into oxidized glutathione (GSSG). It should be noted that GPx can reduce other lipid hydroperoxides as well. There are 5 isoforms of GPx according to different tissue locations. The most abundant isoform, GPx1, is present at the cytoplasmic and mitochondrial level and expressed in most cells.

c.Catalase (CAT)

CAT is a heme enzyme involved in the catalysis of H_2_O_2_ into H_2_O and O_2_ by a dismutation reaction. Of note, the Michaelis–Menten constant (Km) of CAT for H_2_O_2_ is higher than that of GPx, hence the role of GPx is preponderant when H_2_O_2_ concentration is lower. However, the different subcellular localizations of CAT (peroxisome) and GPx (cytosolic) endow them with complementary roles.

### 2.5. Measurement of Oxidative Stress

One of the difficulties in the evaluation of ROS production is their stealth. Only as few methods allow measuring ROS directly. Such methods combine EPR (electron paramagnetic resonance) which is limited to dedicated centers, with experts and the proper equipment, and the fluorescent dyes or FRET sensors such as HyPer Probes, commonly used. These techniques are based on the use of a probe that is weakly fluorescent in its reduced state but becomes highly fluorescent when it is oxidized by ROS with sometime the characteristics to anchor to DNA. Other labs have evaluated the consequences of OS in biological samples, e.g., the presence of oxidized lipids. Indeed, ROS induce lipid peroxidation and the production of markers such as malondialdehyde and isoprostanes [11,34]. Easily quantifiable in plasma, urine, and tissues, they can also be monitored in cell extracts. 8-Hydroxyguanine is a marker of OS produced after ROS reacts with nucleic acids. These markers are often the result of insults to tissue which induce subsequent alterations of membrane lipids, proteins, and nucleic acids, thereby occasioning severe damage up to necrosis and cell death. As previously reported, there are also several molecular couples involved in redox status such as GSH-GSSG, the main redox buffer present at high levels in the cell and regulated by different enzymes including glutathione reductase [11,35]. Other enzymes, such as catalase or the superoxide dismutase, limit the production of ROS. Beyond the regulation of antioxidant enzyme activity, their expression is controlled by gene-related mechanisms, including the NRF2 pathway, which is activated in the case of OS [27,32]. In these conditions, the NRF2-Keap-1 heterodimer is disassembled and NRF2 is translocated to the nucleus, where it acts as a transcription factor for the antioxidant response elements (ARE) of target antioxidant enzymes (e.g., SOD). As a result, redox status can also be evaluated by the levels of these anti-oxidative derivatives.

### 2.6. Preliminary Concepts for Oxidative Stress Models

Importantly, oxidative damage is known to be associated with many physiological or pathological conditions, from aging to atherosclerosis, carcinogenesis, neurodegenerative disorders, and IRI. In this view, the origin and the expansion of OS differ depending on the context and adapted models to evaluate redox status are needed. As mentioned previously, direct determination of ROS is difficult and markers from oxidized macromolecules are mainly used. However, the adapted marker is dependent on cellular and animal models, which present their own limits.

Numerous pharmacological products have been developed to reduce OS development and limit its repercussions. Some of them have shown excellent results in vitro but failed to induce protective effects *in vivo*. The main issue *in vivo* is the high and multiple doses necessary to induce significant results. To limit this issue, some authors have injected molecules *in vivo* that by means of biological degradation form and release, anti-oxidative derivatives over long periods of time. This strategy has been tested in our laboratory with tannic acid, which protects from IRI [25]. Other vectors or an osmotic pump facilitate the delivery and prolongation of the effect over several days [26,27].

In animal models, systemic response helps to evaluate an integrative effect of a target molecule, but the variability of individual response, as well as potentially different genetic backgrounds, may complicate analyses. On the contrary, in a pro-inflammatory context, it seems difficult to evaluate OS induced by inflammatory cells in vitro without the surrounding tissue which could decrease or amplify this response. Indeed, OS is often dependent on a futile reactive sequence.

## 3. Models of Oxidative Stress

The studied models are distinguished by their physiological relevance compared to the human body as observed with large animal models, and their specific targets related to scientific issues as observed with in vitro models (Scheme 3).

### 3.1. Animal Models of Oxidative Stress

Several animal models have been tested or developed to study oxidative stress. In the ‘endogenous’ approach, specific targets of either the pro-oxidant or the anti-oxidant system are genetically modified to study particular aspects of OS pathology. This enables the mimicking of genetic ROS-related diseases and testing of the targeting capability of an anti-oxidant agent [28].

The ‘exogenous’ approach typically involves an animal subjected to a pro-oxidant compound. For instance:-*Oxidative injuries specific to selected organs*. For instance, in the eurotoxin 6-OHDA animal model, the compound is infused within the brain’s ventricular system. This induces depletion of the striatal dopamine, which in turn fosters the production of ROS, injuring neurons [36]. To target the gut, pure ethanol can be used to induce mucosal damage, fostering superoxide anion formation, lipid peroxidation, extracellular matrix degradation, and mitochondrial damage [37].-*OS as a component of diabetes.* It can be explored in alloxan-treated rodents. Alloxan reacts with disulfide bounds, of which the regulation involves the generation of H_2_O_2_. This also produces dialuric acid, further reacting with alloxan to generate ROS and cell death [38]. Other models include Streptozotocin-treated animals, specifically targeting β cells, inducing the depletion of cellular NAD+ and ATP, and promoting xanthine oxidase activation and ROS production.-*Systemic OS.* This can be obtained with *tert*-Butyl hydroperoxide (tBuOOH) [39]. Other approaches use hyperlipidemia, highlighted by an unhealthy diet with high fat (butter, cholesterol, etc.) and subsequent increased plasmatic total and LDL cholesterol levels. Our team determined that LDL oxidation could play a major role in IRI development. Indeed, a high level of LDL oxidation in a large animal model of kidney transplantation was shown to promote severe chronic injury, evidenced through interstitial fibrosis development [40], likely involving OxLDL-induced maladapted vascular repair [41].

Several studies have shown the protective potential of anti-oxidative therapeutics for IR injury [26,42]. However, OS is only one aspect of IR, especially in the context of organ transplantation. The full picture of this syndrome is slowly being unraveled, especially with large animal models [43], which offer the critical advantage of being closer to humans in terms of anatomy and physiology, thereby increasing the translational value of related observations. However, the inherent complexity of their exploitation substantially decreases the number of controllable parameters. So it is that in animal research, whether OS or complex pathologies such as IR are being addressed, it is important when attempting to validate hypotheses to consider the balance between controllability and tranversality. (Scheme 3).

### 3.2. Cellular Models of Oxidative Stress

*In vivo* models are expensive and logistically heavy, display reproducibility issues, and do not facilitate mechanism characterization. Furthermore, ethical recommendations need to be taken into account as formulated by the “3R’s” rules. That is why, in vitro models, including primary cells or cell line culture, represent an interesting alternative to study OS (and hypoxia in particular) [44]. Cellular systems have the advantages of being cheap, flexible, modular, reproducible, compatible with high-throughput screening, and particularly interesting for cell mechanical studies, without systemic interferences. Numerous biological in vitro assays have been developed to evaluate ROS production and cellular antioxidant capacities. To determine ROS at the cellular level, fluorescence microscopy, electron spin resonance (ESR; also called electron paramagnetic resonance), mass spectroscopy and chemical/immunological assays (DHE, TBARS, MDA, 4HNE, 8-*iso*prostane, nitrotyrosine tests, etc.) are established and valuable methods. The measurements of level or activity of SOD, catalase, GSH, and metal chelating assays are used to determine antioxidant status. In addition, the oxygen radical absorbance capacity (ORAC) assay is an accepted method for high-throughput screening of anti-oxidative and/or radical scavenging capacities of compounds. In addition, to determine the intracellular influence of a compound, the cell-based antioxidant activity (CAA) assay can be used, and the ORAC assay provides high-throughput screening of anti-oxidative and/or radical scavenging capacities of compounds [45].

However, the culture of primary cells is limited to a few passages and architecture/polarity/phenotype switch can rapidly occur. Conversely, while immortalized cell lines are stable, their physiological relevance is questioned due to their phenotype modification (allowing extensive proliferation) [46].

Importantly, cellular models in general are suitable for the precise modulation of specific targets via various strategies such as genome editing technologies (CRISPR/Cas9) [47], siRNA, or the use of recombinant protein or protein inhibitors, thereby facilitating study of responses to various stimuli including ROS/anti-ROS balance and hypoxia reoxygenation (HR).

#### Hypoxia-Reoxygenation Models

Cellular models of HR have provided useful tools for OS study [48,49]. Cellular HR appears to be a key signal activating protein regulators, including NRF2 and HIF-1 (Hypoxia Inducible Factor–1); when linked to redox-sensitive antioxidant response, it induces gene expression [48]. To characterize HR disorders, the most optimal in vitro model mimicking oxygen deprivation is an airtight hypoxia chamber [50] or incubator, coupled with controlled temperature and pressure. Nevertheless, few laboratories have access to a hypoxia chamber (or, at least, a CO_2_ incubator with a regulated level of oxygen) to carry out such experiments. One alternative is the use of cobalt chloride (CoCl_2_) in cell culture. CoCl_2_ stabilizes HIF-1α under normoxic conditions, mimicking parts of cellular hypoxic signaling [51]. However, incomplete or ongoing oxygen diffusion/distribution in cell cultures leads to pericellular oxygen levels that are insufficient for cellular metabolic processes. The precise control of pericellular O_2_ levels is required for the appropriate interpretation of results [44]. In addition, the physical condition of cell culture perfusion/superfusion is an important aspect to consider when mimicking physiological oxygen distribution and ROS signaling. For instance, shear stress influences the occurrence of OS [52]. In addition, the assessment of OS and the nature and production level of ROS in hypoxic conditions is dependent on: (i) the hypoxia-inducing compound or condition, (ii) temperature (hypothermia vs. normothermia), (iii) duration of hypoxia, (iv) cell type sensitivity to pO_2_, (v) and composition of the culture/preservation medium.

The interests and limits of the different models are summarized in Table 3.

### 3.3. Ex Vivo Models

*Ex vivo* models are not developed in this review given the pathophysiological mechanisms involved in IRI and oxidative stress. *Ex vivo* evaluation allows studies to be performed without having to have a large number of animals represented. It is done by taking tissue or organ from the animal to perform pharmacological evaluation or to test against the biomaterial outside of the body. This reductionist approach offers some advantages over *in vivo* experiments, including more precise control of experimental conditions and the ability to identify tissue response to specific stimuli. In IRI studies, evaluation of functional parameters is restricted to the early period of reperfusion. The percentage of articles in PubMed published in 2018 that contain the keyword “*ex vivo*” was 0.4% [53]. Furthermore, *ex vivo* models are used for musculoskeletal tissue, including bone, cartilage, muscle, tendon, intervertebral disc, as well as whole diarthrodial joints [53]. *Ex vivo* evaluations of biomaterials are another way for ensuring the safety and efficacy of biomaterials. With regard to biomaterial evaluation, *ex vivo* tests are performed to study the physical, chemical, mechanical, and other properties of biomaterials that have interacted with tissue after being implanted for a time or during the interaction. These models also have a new interest for transplantation and organ preservation, taking into account the development of machine preservation and normothermic perfusion, particularly for lung, kidney, and liver [54,55]. In addition, new developments of *ex vivo* can be expected with cellular bioengineering methods (organotypic cultures based on 3D cultures of primary cells, organoids and “organ-on chips” technology, and genetically modified animals).

## 4. New Concepts in Oxidative Stress

### 4.1. Cellular Bioengineering

As stated above, all fundamental and translational research models have their limits. There is an unmet need to develop new, more accurate, more realistic and robust experimentation models that recapitulate the cellular microenvironment. Evolving towards alternative and intermediary models is crucial and different strategies are emerging in cellular bioengineering leading to the generation of human biological and multicellular structures/tissues, which are more and more complex and organized. In recent years, interest in different 3D culture techniques has increased since they have been shown to be more relevant, reflecting human tissues and disease conditions. One of these emerging techniques is 3D bioprinting, which is a precise technique that facilitates control of the spatial cell distribution through layer-by-layer assembly in a 3D pattern. Bioprinting is able to generate 3D tissues, which are analogous to complexly organized human tissues, through precise spatial positioning of materials and cells. Organoids derived from human-induced pluripotent stem cells (hiPSCs) are one great example of such innovative technologies (schema 4). Indeed, cellular reprogramming through the induction of pluripotent stem cells from somatic stem cells opens the door to many applications, including in fundamental research [56]. Protocols to generate organoids from hiPSCs are now published for multiple organs including kidney, liver, brain, heart, gut, retina, etc. One major advantage of this technology is the possibility to choose the cell’s genetic background, either by selecting individuals with a specific genetic disease or by using genome editing technologies such as CRISR/Cas9 to perform targeted gene modification, including genes of interest in OS control [57,58,59].

The interests and limits of organoids are summarized in Scheme 4.

Organoids are particularly appealing in attempts to reproduce pathophysiological conditions and the complexity of cellular interactions in different fields: developmental study, disease modeling, cancer research, toxicology/pharmacology screening, etc... Regarding OS, some examples have been reported in the literature. Vergara et al. detected significant ROS production (using DHE) in retina organoids treated for 3 h with increasing concentrations of hydrogen peroxide [60]. In cancer research, results from the SELECT trial were recapitulated in vitro using one alternative 3-dimensional (3D) prostate organoid culture, arguing that an organoid model could increase the predictive value of in vitro studies for *in vivo* outcomes [61]. The SELECT clinical trial (NCT00006392) showed that selenium was not efficient in reducing prostate cancer incidence, while vitamin E was associated with increased risk of the disease.

Aside from hiPSC-derived cells, organotypic cultures based on 3D cultures of primary cells are of interest in OS studies. For instance, esophageal 3D organotypic culture methods using epithelial cells grown on top of collagen/Matrigel matrices containing human fetal esophageal fibroblasts are being used to study chronic inflammatory conditions of the gastrointestinal tracts including OS and associated DNA damage [62,63].

Moreover, these structures can be combined with microfluidics, leading to “organ-on-chip” technology; for example, Hale et al. developed a “glomerular chip” containing layered hiPSC-derived mature podocytes that structurally and functionally mimic the kidney glomerular membrane [64]. We can expect that such structures will be able to recapitulate cell mechanisms (related to various cells and tissues) linked with different kinds of stresses such as shear stress or OS.

Finally, one could also expect interesting results from bioprinting approaches, which allow on-demand generation of pieces of cell sheets and tissues, with flexible and precise time and dimension control [65]. Based on the large panel of available bioinks, bioprinting opens the door to a large number of possibilities recapitulating complex microenvironments, including precise regulation of the composition of various cytokines, inflammatory protein, and OS-linked species. For instance, 3D bioprinting is expected to recapitulate cancer microenvironments based on its ability to precisely define perfusable networks and the positions of different cell types [66].

### 4.2. Molecular Modelling

A promising way to study OS and antioxidant action is by using molecular modeling techniques, i.e., quantum chemistry calculation (mainly based on DFT—density functional theory) or molecular dynamic (MD) simulations (Scheme 5). These *in silico* methods have become particularly robust and their growing use has paved the way towards efficient design of antioxidants. We previously highlighted in this review the fact that antioxidants act through different mechanisms including free radical scavenging, chelation of transition metal ions to form inert complexes, inhibition of lipid peroxidation, or inhibition of enzymes producing ROS. In this section, we briefly describe how these mechanisms can be studied *in silico*. Other indirect antioxidant effects can be studied by molecular modeling, including DNA repair or anti-inflammatory activity, but their description is beyond the scope of this section.

*Free radical scavenging*—Over the past two decades, a plethora of studies have used DFT calculations to predict or confirm free radical scavenging by antioxidants, mainly containing OH or NH groups [68,69]. DFT formalism is particularly adapted to evaluation of such activity, provided that proper DFT-functionals are used (e.g., hybrid or *meta*GGA functionals) and that basis sets are large enough to ensure correct description of the electronic orbitals (e.g., 6-31+G(d,p) or higher). Various calculated descriptors accurately describe this activity. Among them, bond dissociation enthalpy (BDE; H°_(298K)_(AntiOx^●^) + H°_(298K)_(H^●^)–H°_(298K)_(AntiOx-H)) is the major descriptor for many antioxidants [70,71,72], which predict DPPH scavenging activity with high robustness. When dealing with other antioxidant assays (e.g., ABTS^+•^, ORAC, electrochemistry), other descriptors can be calculated to better interpret activity, including energies and distributions of frontier orbitals (HOMO—highest occupied molecular orbitals, and LUMO—lowest unoccupied molecular orbitals), ionization potential, spin and electron densities, deprotonation energies [73,74,75]. The robustness of these estimations is ensured by the fact that quantum chemistry perfectly catches the underlying mechanism of free radical scavenging, mainly H-atom transfer, electron transfer, or adduct formation between the antioxidant and the free radical. Beyond thermodynamic descriptors, kinetic parameters can drive competition between the different mechanisms in biological environments. DFT-based studies have evaluated the rate constants of free radical scavenging by phenolic compounds with reasonable agreement compared to experimental values [76,77,78,79]. In DFT calculations, solvent effects can also be considered by using implicit solvent models or by including a few explicit solvent molecules in the calculations (at least in the first solvent shell) to account for specific interaction between the antioxidant and the solvent (e.g., H-bonds)

*Metal chelation*—The formation of ROS is often mediated by metals, either in (heme or non-heme) metallo-enzymes or by free metals in certain compartments such as the stomach. Various *in silico* studies have rationalized the binding of antioxidants into the active site of metal ion -containing enzymes [80], while less theoretical studies have directly calculated metal-antioxidant binding in aqueous solution [81]. This can be done by using DFT calculations and adapted basis set to describe the electron orbitals of the metal. Additionally, the different charge and spin states of the antioxidant-metal complexes should be systematically considered [69,82].

*Lipid peroxidation inhibition*—To efficiently inhibit the oxidation of membrane lipids, antioxidants should efficiently block the propagation stage of this process. For that purpose, the compounds should not only efficiently scavenge free radicals but also be inserted sufficiently deeply in the lipid bilayer. MD simulations can calculate the capacity of antioxidants to be inserted into lipid bilayers and, when available, to indicate their position and orientation in membranes with remarkable accuracy compared to experimental data [27,83,84,85]. Many antioxidants rapidly approach the lipid bilayers from the bulk water, and they partition in the membrane at different depths of insertion and orientation depending, to a major extent, on their lipophilic/hydrophilic/amphiphilic character. Other parameters should be considered including lipid composition, lipid phase, and the presence of permeability-enhancers or antioxidant concentration. The atomic picture provided by MD simulations provides atomistic-resolved control of the depth of penetration in the membranes. Knowing the precise positioning of antioxidants in membranes supports the design of new lipid peroxidation inhibitors, which can be anchored to the membrane at a well-defined depth to optimize inhibition of the propagation stage of lipid peroxidation. Interestingly, MD simulations have highlighted noncovalent interactions between various antioxidants (e.g., vitamin E, vitamin C, and polyphenols) inside the membranes, favoring synergism between antioxidants by regeneration processes [67]. The possibility of synergisms in such noncovalent antioxidant aggregates and the action of antioxidants according to different mechanisms is impelling researchers towards well-defined antioxidant cocktails, favoring multi-action and synergism [86].

*Oxidative enzyme inhibition*—Ligand-protein can also be efficiently studied by molecular modeling, for example, in study of oxidative enzyme inhibition. Numerous docking studies have been conducted to understand the mode of binding and the mechanism of xanthine oxidase inhibition. These *in silico* studies have established structure-activity relationships and elucidation of the driving forces responsible for the stability of the ligand-xanthine oxidase complexes [87,88,89,90]. Many polyphenols have been shown to establish H-bond and van der Waals interactions in a cavity containing hydrophobic pockets. Comprehension of non-specific and specific interactions has furthered proposals of chemical modifications aimed at strengthening binding and increasing xanthine oxidase inhibition activity [91,92]. Various flavonoids or other derivatives appear efficient at forming stable complexes with xanthine oxidase due to their amphiphilic character favoring hydrophobic contacts and specific H-bonds with their planar π-conjugated systems and their OH groups, respectively [87].

### 4.3. Mathematical Modeling at the Cell Level

The previous section illustrates how *molecular* modeling provides better understanding of redox reactions and intrinsic mechanisms. However, this approach ignores cellular heterogeneity and higher-level processes, as they occur in biological compartments. The cell is an integrated system, and multi-level processes modulate its behavior and fate in response to redox events, whether they be pathological or physiological. Excess ROS can be variably deleterious to cellular components and compartments. On the other hand, ROS also plays a decisive role in cell and subcellular regulations, including mitochondrial function, metabolism, and signaling. These events are complex and difficult to analyze experimentally. As such, they are often overlooked or overly simplified (e.g., “oxidative stress”). In order to develop experimental models of OS that are physiopathologically and therapeutically relevant, it is necessary to understand cellular redox biology in integrated constructs, qualitatively and quantitatively [93,94,95]. Here, we briefly illustrate cell redox complexity and survey existing modeling and simulation (MS) attempts at the cellular level.

#### 4.3.1. Overview of Cell Redox Complexity Warranting Mathematical Modeling Combined with Quantitative Approaches.

Reactions are thermodynamically driven (favored or hampered) by the redox potentials of the reacting couples, the latter depending on numerous factors, including pH, temperature, cell compartment, and local concentrations (actually, activities). On the other hand, the kinetic properties of these reactions (their actual rates), enzyme-catalyzed or not, are driven by the same factors. Furthermore, reactions can be either diffusion-limited or reaction-limited, warranting the integration of both kinetic and thermodynamic equations for the processes under scrutiny [96]. Redox reactions are strongly compartmentalized, and the relevant local levels of controlling factors are often uncertain. The number of redox species is considerable, and the chemistry of their reactions is tightly intricate: any “reactive species” will drive the production of one or many others. One textbook example is the production of the hydroxyl radical (OH°) from superoxide anion (O_2_°^−^) and hydrogen peroxide (H_2_O_2_) by the so-called iron-catalyzed Haber–Weiss reaction [97]. Another key example is the production, from O_2_°^−^ and nitric oxide (NO^°^), of the peroxynitrite anion (ONOO^−^). Moreover, reactive species can modulate the activity and/or function of numerous enzymes and drive adaptive gene expression. For instance, redox metabolism is tightly and reciprocally modulated by energetic metabolism [98]. Finally, different cell types usually have different redox functionalities and capacities, such as different response ranges against OS [99]. To quote a recent review dedicated to redox systems biology: “Systems biology investigates multiple components in complex environments and can provide integrative insights into the multifaceted cellular redox state” [29], cell redox complexity hampers quantitative integration and challenges qualitative reasoning. These considerations have impelled biochemists and cell biologists to turn to mathematical modeling and computer simulation (MS) as a means of empowering the experimental method and of fostering analysis and hypothesis testing [94]. For the interested reader, we provide hereafter a comparative summary of the main MS formalisms and associated tools: equation-based modeling (EBM), agent-based modeling (ABM), and logic-based modeling (see Table 4). We briefly present the methodology and provide both an introductory paper as well as an application example related to the field of oxidative stress (if available). EBM is based on kinetic equations for describing reaction rates and dynamic equations for the integration of the corresponding state variables [100]. Using EBM and sophisticated enzyme kinetics, the study by Benfeitas et al. was able to quantitatively reproduce and analyze Prx2 regimen of peroxide elimination in human erythrocytes [101]. ABM is based on individual (or autonomous) agents (e.g., mitochondria) interacting with their environment, including other agents [102]. In the field of OS, Park et al., addressed the influence of mitochondrial network dynamics on intracellular ROS propagation [103]. Finally, LBM is based on “logical rules” to calculate the attractors (“steady-state”) and the evolution dynamic of an interaction network of discrete components, typically gene expression or transduction signaling pathways [104].

#### 4.3.2. A Brief Survey of Mathematical Modelling and Simulation of Cell Redox Biology

In the 1980s–1990s, very few MS approaches were developed, partly because the development of powerful personal computers and convenient mathematical software was only starting to emerge. The landmark work is undoubtedly the integrative kinetic model of lipid peroxidation in mitochondrial inner membrane, developed by Antunes et al. [105]. In this impressively detailed MS study, the authors describe and integrate more than 40 redox chemical and enzymatic reactions (58 ODE’s) in the membrane lipid phase and the aqueous compartment. In addition, the authors investigated peroxidation of saturated versus unsaturated lipids and implemented reactions including the anti- and pro-oxidant effects of vitamin E, the pro-oxidant effects of iron, the action of phospholipase A2, glutathione-dependent peroxidases, glutathione reductase and superoxide dismutase, production of O_2_°^−^ radical by the respiratory chain, as well as oxidative damage to proteins and DNA [105]. Under current standards, the main limitation of this remarkable work is its partial validation, most likely due to the ambitious biochemical scope and the limited experimental data available at the time, associated with a lack of thermodynamic constraints. Given the importance of lipid peroxidation in IR injuries, this model would benefit from updating and re-implementation with modern MS tools.

Based on cardiomyocyte electrophysiological models developed in the 1980s [106], numerous MS studies have targeted the involvement of mitochondria in redox biology and signaling in heart cells, such as experimentally demonstrated ROS oscillations and ROS-induced ROS release [107,108,109]. Of importance to normal mitochondrial respiration as well as IRI, ROS production by the mitochondrial respiratory chain has been modeled and validated [110]. In the last two decades, due to the specific characteristics and importance of the –SH group in cell redox status, a growing number of MS studies have targeted thiol-based systems (e.g., GSH, thioredoxin, peroxiredoxin) [30,31], and the “cysteine redoxome” concept has been coined [29]. Most recently, in order to both quantitatively and dynamically reproduce dedicated or bibliographical experimental data, MS studies have implemented redox processes, in multi-compartment/multi-dynamic approaches, such as: production and elimination kinetics of H_2_O_2_ [105,111], coupling between redox metabolism and energetic metabolism [107,112,113]; quantitative and MS analysis of the modulation of cell signaling by redox processes [114]; MS description of redox processes and metabolism in single cells, such as the erythrocyte [115] or micro-organisms [116]. Higher integrated levels of ROS and redox-related pathophysiological issues are also addressed through the MS paradigm [93]. Examples include the role of ROS in the renal medulla [117] and the involvement of the O_2_°^−^ radical cisplatin-induced kidney dysfunction [118]; in the liver associated with IRI [119]; and in neurodegenerative diseases [97] and cancer [120].

## 5. Conclusions

OS is a known established component of IR injury. The massive generation of mitochondrial ROS contributes to a deleterious cascade of IR-induced events leading to severe cellular damage. Up until now, despite frequent in vitro efficiency, potential antioxidant therapies have often failed to be successfully translated into clinical practice. Therefore, the unraveling of such complex phenomena warrants the development of new approaches, alone or in combination, such as microfluidics, organoids, and *in silico* modeling.

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
