# Peer review of "Oxidative Stress Evaluation in Ischemia Reperfusion Models: Characteristics, Limits and Perspectives"

_ijms, 2021, doi:10.3390/ijms22052366_

Round 1

Reviewer 1 Report

The manuscript “Oxidative Stress evaluation in ischemia reperfusion models: Characteristics, Limits and Perspectives” by Chazelas et al. has improved significantly. Especially the addition of the tables has added value to the manuscript. I still have two minor comments.

The authors addressed all my concerns except for the English review. While the language of the manuscript has been improved significantly there are still passages that are difficult to understand (e.g. page 7 line 214 Such methods combine EPR (Electron Paramagnetic Resonance) which is limited to dedicated centers, with experts and the proper equipment and the fluorescent dyes or FRET sensors such as HyPer Probes, commonly used. This sentence would imply that “such methods” combine EPR with experts and proper equipment. I assume there is a part missing). Please don’t correct just this sentence but rigorously correct the manuscript in whole. There are a lot of typos that make reading sometimes difficult.

Please also review your subheadings and correct them where appropriate. There are some subheadings that do not reflect the following text. (e.g. page 8 line 286 Systemic OS can be obtained with tert-Butyl hydroperoxide (tBuOOH) with the following text describing effects of lipids and unhealthy diet.) Again, while most headings are adequate, this is only one example. Please review the entire manuscript.

Author Response

We have made the modifications required. You will find enclosed the revision of our manuscript entitled "Oxidative Stress Models: Characteristics, Limitations and Perspectives". We would like to thank you for your constructive and helpful comments. More specific answers are provided in the corrected document .

Reviewer 2 Report

The authours have addressed all minor issues.

From my perspective, this is an interesting review for all researchers that study the oxidative stress injury in different fields (from transplantation to shock hypoxic damage). This review suggests new models to improve our experiments in the field of ischemia/reperfusion injury. 
I believe that this review should be of broad interest to the readership of the international Journal of molecular sciences.

Author Response

You will find enclosed the revision of our manuscript entitled "Oxidative Stress Models: Characteristics, Limitations and Perspectives". We would like to thank you for your constructive and helpful comments. More specific answers are provided in the corrected document .

This manuscript is a resubmission of an earlier submission. The following is a list of the peer review reports and author responses from that submission.

Round 1

Reviewer 1 Report

The review Models of Oxidative Stress: Characteristics, Limits and Perspectives by Chazelas et al aims at providing a comprehensive overview of ROS from chemistry to biology. This is an extremely ambitioned endeavor given the vast range and scope of the subject. I very mucg like the approach of connecting chemical behavior to biological functionality that the authors outline in the abstract and I’m certain that the field could benefit significantly from this work.

Major issues:

1) However, the extremely high-set aim is also the main downfall of this review. Because the authors wanted to cover such a big field, they rarely can provide insights beyond mere stating what the literature said. The authors did not make any connections between the facts they report, nor did they discuss or at least outline conflicting reports from the literature.

2) The reason for focusing on ischemia and reperfusion escapes me. If the idea was to limit the scope as to avoid comment 1, they failed. Maybe the authors instead of focusing on IRI could focus on a biological entity e.g. ROS on the plasma membrane or ROS in the mitochondrion. This could potentially focus the approach and make this work significantly better.

3) Please but a lot more work into developing a coherent storyline – this would increase readability a lot.

4) I accept that this might be a personal preference, but I would advise the authors to include more figures. A figure can introduce and explain the complicated concepts of ROS generation, antioxidant control, diffusion and signaling a lot easier than words, which would increase understanding and citation index for this article. This is not a must, but an advice.

5) Having worked with ROS myself, I fail to see why fluorescent (or FRET) based probes are almost missing in the description (except for one or two sentences in 184-186). HyPer Probes for example are recognized almost universally as gold standard in live cell ROS visualization.

6) Please vigorously correct your English. Some parts (especially the first part) are very well written, but some paragraphs are almost incomprehensible. This significantly blunts the impact of this article because the common reader will not fight through these paragraphs.

7) Please also take some time to review your references. I have only checked a couple but at least once – according to the abstract – it covered a different topic than the one it was used to reference. Please also don’t reference reviews in French (even if it is yours) when you could reference the original work.

I have a few minor comments as well:

Line 82: Please remove the comment by “Pat”. Also, I agree phenomenon would be the better word.

Line 83: To avoid misunderstandings I would suggest describing ROS as oxidized or (partially) oxidized molecular species. While the term partially reduced may technically be true, the matter of the fact is that ROS is the oxidizing agent and as such needs to be in an oxidized state. Your way of describing it would spark confusion with a biologist trying to separate ROS from the antioxidant, reducing factors, which are in fact reduced.

Line 96: The introduction of free radicals comes rather abrupt. Also, I’m not sure what this introduction is doing there. This sentence implies that ROS would be radicals which you yourself retract two sentences further in, please remove.

Line 176: Your statement The main problem with evaluating OS is the stealth appearance of ROS. The only adapted technic is the EPR (Electron Paramagnetic Resonance) but this method is limited to dedicated centers, with experts and the proper equipment. should be rewritten. It implies that there is no way to measure ROS except for EPR, which is not true. One example would be the fluorescent dyes (or FRET sensors) mentioned above. If you mean direct measurement of ROS, which the rest of the paragraph implies, please write so. Also, I don’t see the problem there, since many measurements in biology are indirect (i.e. protein detection by antibodies via fluorescent dyes). I don’t understand why that needs to be pointed out.

Reviewer 2 Report

The review is rich, well written and well organised.

First of all the manuscript is rich, well written and organized. They authors propose an exhaustive overview, going through the description of the principal mechanisms of I/R and oxidative stress injury (biochemical point of view); the different methods (to our knowledge), in vitro and in vivo, used to better represent the oxidative stress condition; the advantages and limits of these procedures, which often did not perfectly mimic ischemia damage; and finally describing new systems (as 3D organoids and in silico models) to overcome the limits of the "old" procedures.
From my perspective, this is an interesting review for all researchers that study the oxidative stress injury in different fields (from transplantation to shock hypoxic damage). This review suggests new models to improve our experiments in the field of ischemia/reperfusion injury.
I believe that this review should be of broad interest to the readership of the international Journal of molecular sciences.

in Figure 1: the lettering in the figure should be in bold as it is not very visible.